# Beta-Negative Binomial Process and Exchangeable Random Partitions for Mixed-Membership Modeling

**Mingyuan Zhou**
IROM Department, McCombs School of Business
The University of Texas at Austin, Austin, TX 78712, USA
`mingyuan.zhou@mccombs.utexas.edu`

## Abstract

The beta-negative binomial process (BNBP), an integer-valued stochastic process, is employed to partition a count vector into a latent random count matrix. As the marginal probability distribution of the BNBP that governs the exchangeable random partitions of grouped data has not yet been developed, current inference for the BNBP has to truncate the number of atoms of the beta process. This paper introduces an exchangeable partition probability function to explicitly describe how the BNBP clusters the data points of each group into a random number of exchangeable partitions, which are shared across all the groups. A fully collapsed Gibbs sampler is developed for the BNBP, leading to a novel nonparametric Bayesian topic model that is distinct from existing ones, with simple implementation, fast convergence, good mixing, and state-of-the-art predictive performance.

## 1 Introduction

For mixture modeling, there is a wide selection of nonparametric Bayesian priors, such as the Dirichlet process [1] and the more general family of normalized random measures with independent increments (NRMIs) [2, 3]. Although a draw from an NRMI usually consists of countably infinite atoms that are impossible to instantiate in practice, one may transform the infinite-dimensional problem into a finite one by marginalizing out the NRMI. For instance, it is well known that the marginalization of the Dirichlet process random probability measure under multinomial sampling leads to the Chinese restaurant process [4, 5]. The general structure of the Chinese restaurant process is broadened by [5] to the so called exchangeable partition probability function (EPPF) model, leading to fully collapsed inference and providing a unified view of the characteristics of various nonparametric Bayesian mixture-modeling priors. Despite significant progress on EPPF models in the past decade, their use in mixture modeling (clustering) is usually limited to a single set of data points.

Moving beyond mixture modeling of a single set, there has been significant recent interest in mixed-membership modeling, $i.e.$, mixture modeling of grouped data $\boldsymbol{x}_1, \ldots, \boldsymbol{x}_J$, where each group $\boldsymbol{x}_j = \{x_{ji}\}_{i=1, m_j}$ consists of $m_j$ data points that are exchangeable within the group. To cluster the $m_j$ data points in each group into a random, potentially unbounded number of partitions, which are exchangeable and shared across all the groups, is a much more challenging statistical problem. While the hierarchical Dirichlet process (HDP) [6] is a popular choice, it is shown in [7] that a wide variety of integer-valued stochastic processes, including the gamma-Poisson process [8, 9], beta-negative binomial process (BNBP) [10, 11], and gamma-negative binomial process (GNBP), can all be applied to mixed-membership modeling. However, none of these stochastic processes are able to describe their marginal distributions that govern the exchangeable random partitions of grouped data. Without these marginal distributions, the HDP exploits an alternative representation known as the Chinese restaurant franchise [6] to derive collapsed inference, while fully collapsed inference is available for neither the BNBP nor the GNBP.

The EPPF provides a unified treatment to mixture modeling, but there is hardly a unified treatment to mixed-membership modeling. As the first step to fill that gap, this paper thoroughly investigates the law of the BNBP that governs its exchangeable random partitions of grouped data. As directly deriving the BNBP's EPPF for mixed-membership modeling is difficult, we first randomize the group sizes $\{m_j\}_j$ and derive the joint distribution of $\{m_j\}_j$ and their random partitions on a shared list of exchangeable clusters; we then derive the marginal distribution of the group-size count vector $\boldsymbol{m} = (m_1, \ldots, m_J)^T$, and use Bayes' rule to further arrive at the BNBP's EPPF that describes the prior distribution of a latent column-exchangeable random count matrix, whose $j$th row sums to $m_j$.

The general method to arrive at an EPPF for mixed-membership modeling using an integer-valued stochastic process is an important contribution. We make several additional contributions: 1) We derive a prediction rule for the BNBP to simulate exchangeable random partitions of grouped data governed by its EPPF. 2) We construct a BNBP topic model, derive a fully collapsed Gibbs sampler that analytically marginalizes out not only the topics and topic weights, but also the infinite-dimensional beta process, and provide closed-form update equations for model parameters. 3) The straightforward to implement BNBP topic model sampling algorithm converges fast, mixes well, and produces state-of-the-art predictive performance with a compact representation of the corpus.

## 1.1 Exchangeable Partition Probability Function

Let $\Pi_m = \{A_1, \ldots, A_l\}$ denote a random partition of the set $[m] = \{1, 2, \ldots, m\}$, where there are $l$ partitions and each element $i \in [m]$ belongs to one and only one set $A_k$ from $\Pi_m$. If $P(\Pi_m = \{A_1, \ldots, A_l\}|m)$ depends only on the number and sizes of the $A_k$'s, regardless of their order, then it is called an exchangeable partition probability function (EPPF) of $\Pi_m$. An EPPF of $\Pi_m$ is an EPPF of $\Pi := (\Pi_1, \Pi_2, \ldots)$ if $P(\Pi_m|n) = P(\Pi_m|m)$ does not depend on $n$, where $P(\Pi_m|n)$ denotes the marginal partition probability for $[m]$ when it is known the sample size is $n$. Such a constraint can also be expressed as an addition rule for the EPPF [5]. In this paper, the addition rule is not required and the proposed EPPF is allowed to be dependent on the group sizes (or sample size if the number of groups is one). Detailed discussions about sample size dependent EPPFs can be found in [12]. We generalize the work of [12] to model the partition of a count vector into a latent column-exchangeable random count matrix. A marginal sampler for $\sigma$-stable Poisson-Kigman mixture models (but not mixed-membership models) is proposed in [13], encompassing a large class of random probability measures and their corresponding EPPFs of $\Pi$. Note that the BNBP is not within that class and both the BNBP's EPPF and perdition rule are dependent on the group sizes.

## 1.2 Beta Process

The beta process $B \sim \mathrm{BP}(c, B_0)$ is a completely random measure defined on the product space $[0, 1] \times \Omega$, with a concentration parameter $c > 0$ and a finite and continuous base measure $B_0$ over a complete separable metric space $\Omega$ [14, 15]. We define the Lévy measure of the beta process as

$$\nu(dpd\omega) = p^{-1}(1 - p)^{c-1}dpB_0(d\omega). \tag{1}$$

A draw from $B \sim \mathrm{BP}(c, B_0)$ can be represented as a countably infinite sum as $B = \sum_{k=1}^{\infty} p_k \delta_{\omega_k}$, $\omega_k \sim g_0$, where $\gamma_0 = B_0(\Omega)$ is the mass parameter and $g_0(d\omega) = B_0(d\omega)/\gamma_0$ is the base distribution. The beta process is unique in that the beta distribution is not infinitely divisible, and its measure on a Borel set $A \subset \Omega$, expressed as $B(A) = \sum_{k:\omega_k \in A} p_k$, could be larger than one and hence clearly not a beta random variable. In this paper we will work with $Q(A) = -\sum_{k:\omega_k \in A} \ln(1 - p_k)$, defined as a logbeta random variable, to analyze model properties and derive closed-form Gibbs sampling update equations. We provide these details in the Appendix.

## 2 Exchangeable Cluster/Partition Probability Functions for the BNBP

The integer-valued beta-negative binomial process (BNBP) is defined as

$$X_j|B \sim \mathrm{NBP}(r_j, B), \ B \sim \mathrm{BP}(c, B_0), \tag{2}$$

where for the $j$th group $r_j$ is the negative binomial dispersion parameter and $X_j|B \sim \mathrm{NBP}(r_j, B)$ is a negative binomial process such that $X_j(A) = \sum_{k:\omega_k \in A} n_{jk}$, $n_{jk} \sim \mathrm{NB}(r_j, p_k)$ for each Borel set $A \subset \Omega$. The negative binomial distribution $n \sim \mathrm{NB}(r, p)$ has probability mass function (PMF) $f_N(n) = \frac{\Gamma(n+r)}{n!\Gamma(r)}p^n(1 - p)^r$ for $n \in \mathbb{Z}$, where $\mathbb{Z} = \{0, 1, \ldots\}$. Our definition of the BNBP follows

those of [10, 7, 11], where for inference [10, 7] used finite truncation and [11] used slice sampling. There are two recent papers [16, 17] that both marginalize out the beta process from the negative binomial process, with the predictive structures of the BNBP described as the negative binomial Indian buffet process (IBP) [16] and "ice cream" buffet process [17], respectively. Both processes are also related to the "multi-scoop" IBP of [10], and they all generalize the binary-valued IBP [18]. Different from these two papers on infinite random count matrices, this paper focuses on generating a latent column-exchangeable random count matrix, each of whose row sums to a fixed observed integer. This paper generalizes the techniques developed in [17, 12] to define an EPPF for mixed-membership modeling and derive truncation-free fully collapsed inference.

The BNBP by nature is an integer-valued stochastic process as $X_j(A)$ is a random count for each Borel set $A \subset \Omega$. As the negative binomial process is also a gamma-Poisson mixture process, we can augment (2) as a beta-gamma-Poisson process as

$$X_j | \Theta_j \sim \mathrm{PP}(\Theta_j), \ \Theta_j | r_j, B \sim \Gamma\mathrm{P}[r_j, B/(1 - B)], \ B \sim \mathrm{BP}(c, B_0),$$

where $X_j | \Theta_j \sim \mathrm{PP}(\Theta_j)$ is a Poisson process such that $X_j(A) \sim \mathrm{Pois}[\Theta_j(A)]$, and $\Theta_j | B \sim \Gamma\mathrm{P}[r_j, B/(1-B)]$ is a gamma process such that $\Theta_j(A) = \sum_{k:\omega_k \in A} \theta_{jk}$, $\theta_{jk} \sim \mathrm{Gamma}[r_j, p_k/(1 - p_k)]$, for each Borel set $A \subset \Omega$. The mixed-membership-modeling potentials of the BNBP become clear under this augmented representation. The Poisson process provides a bridge to link count modeling and mixture modeling [7], since $X_j \sim \mathrm{PP}(\Theta_j)$ can be equivalently generated by first drawing a total random count $m_j := X_j(\Omega) \sim \mathrm{Pois}[\Theta_j(\Omega)]$ and then assigning this random count to disjoint disjoint Borel sets of $\Omega$ using a multinomial distribution.

## 2.1  Exchangeable Cluster Probability Function

In mixed-membership modeling, the size of each group is observed rather being random, thus although the BNBP's augmented representation is instructive, it is still unclear how exactly the integer-valued stochastic process leads to a prior distribution on exchangeable random partitions of grouped data. The first step for us to arrive at such a prior distribution is to build a sample size dependent model that treats the number of data points to be clustered (partitioned) in each group as random. Below we will first derive an exchangeable cluster probability function (ECPF) governed by the BNBP to describe the joint distribution of the random group sizes and their random partitions over a random, potentially unbounded number of exchangeable clusters shared across all the groups. Later we will show how to derive the EPPF from the ECPF using Bayes' rule.

**Lemma 1.** *Denote $\delta_k(z_{ji})$ as a unit point mass at $z_{ji} = k$, $n_{jk} = \sum_{i=1}^{m_j} \delta_k(z_{ji})$, and $X_j(A) = \sum_{k:\omega_k \in A} n_{jk}$ as the number of data points in group $j$ assigned to the atoms within the Borel set $A \subset \Omega$. The $X_j$'s generated via the group size dependent model as*

$$z_{ji} \sim \sum_{k=1}^{\infty} \frac{\theta_{jk}}{\Theta_j(\Omega)} \delta_k, \ m_j \sim \mathrm{Pois}(\Theta_j(\Omega)),$$
$$\Theta_j \sim \Gamma\mathrm{P}[r_j, B/(1 - B)], \ B \sim \mathrm{BP}(c, B_0) \tag{3}$$

*is equivalent in distribution to the $X_j$'s generated from a BNBP as in (2).*

*Proof.* With $B = \sum_{k=1}^{\infty} p_k \delta_{\omega_k}$ and $\Theta_j = \sum_{k=1}^{\infty} \theta_{jk} \delta_{\omega_k}$, the joint distribution of the cluster indices $\boldsymbol{z}_j = (z_{j1}, \ldots, z_{jm_j})$ given $\Theta_j$ and $m_j$ can be expressed as $p(\boldsymbol{z}_j | \Theta_j, m_j) = \prod_{i=1}^{m_j} \frac{\theta_{jz_{ji}}}{\sum_{k'=1}^{\infty} \theta_{jk'}} = \frac{1}{(\sum_{k=1}^{\infty} \theta_{jk})^{m_j}} \prod_{k=1}^{\infty} \theta_{jk}^{n_{jk}}$, which is not in a fully factorized form. As $m_j$ is linked to the total random mass $\Theta_j(\Omega)$ with a Poisson distribution, we have the joint likelihood of $\boldsymbol{z}_j$ and $m_j$ given $\Theta_j$ as

$$f(\boldsymbol{z}_j, m_j | \Theta_j) = f(\boldsymbol{z}_j | \Theta_j, m_j)\mathrm{Pois}(m_j, \Theta_j(\Omega)) = \frac{1}{m_j!} \prod_{k=1}^{\infty} \theta_{jk}^{n_{jk}} e^{-\theta_{jk}}, \tag{4}$$

which is fully factorized and hence amenable to marginalization. Since $\theta_{jk} \sim \mathrm{Gamma}[r_j, p_k/(1 - p_k)]$, we can marginalize $\theta_{jk}$ out analytically as $f(\boldsymbol{z}_j, m_j | r_j, B) = \mathbb{E}_{\Theta_j}[f(\boldsymbol{z}_j, m_j | \Theta_j)]$, leading to

$$f(\boldsymbol{z}_j, m_j | r_j, B) \ \ = \frac{1}{m_j!} \prod_{k=1}^{\infty} \frac{\Gamma(n_{jk} + r_j)}{\Gamma(r_j)} p_k^{n_{jk}} (1 - p_k)^{r_j}. \tag{5}$$

Multiplying the above equation with a multinomial coefficient transforms the prior distribution for the categorical random variables $\boldsymbol{z}_j$ to the prior distribution for a random count vector as $f(n_{j1}, \ldots, n_{j\infty} | r_j, B) = \frac{m_j!}{\prod_{k=1}^{\infty} n_{jk}!} f(\boldsymbol{z}_j, m_j | r_j, B) = \prod_{k=1}^{\infty} \mathrm{NB}(n_{jk}; r_j, p_k)$. Thus in the prior,

for each group, the sample size dependent model in ( 3) draws $n_{jk} \sim \mathrm{NB}(r_j, p_k)$ random number of data points independently at each atom. With $X_j := \sum_{k=1}^{\infty} n_{jk}\delta_{\omega_k}$, we have $X_j|B \sim \mathrm{NBP}(r_j, B)$ such that $X_j(A) = \sum_{k:\omega_k \in A} n_{jk}$, $n_{jk} \sim \mathrm{NB}(r_j, p_k)$. $\qquad\square$

The Lemma below provides a finite-dimensional distribution obtained by marginalizing out the infinite-dimensional beta process from the BNBP. The proof is provided in the Appendix.

**Lemma 2** (ECPF). *The exchangeable cluster probability function (ECPF) of the BNBP, which describes the joint distribution of the random count vector $\boldsymbol{m} := (m_1, \ldots, m_J)^T$ and its exchangeable random partitions $\boldsymbol{z} = (z_{11}, \ldots, z_{Jm_J})$, can be expressed as*

$$f(\boldsymbol{z}, \boldsymbol{m}|\boldsymbol{r}, \gamma_0, c) = \frac{\gamma_0^{K_J} e^{-\gamma_0[\psi(c+r.)-\psi(c)]}}{\prod_{j=1}^{J} m_j!} \prod_{k=1}^{K_J} \left[ \frac{\Gamma(n_{\cdot k})\Gamma(c+r.)}{\Gamma(c+n_{\cdot k}+r.)} \prod_{j=1}^{J} \frac{\Gamma(n_{jk}+r_j)}{\Gamma(r_j)} \right], \qquad (6)$$

*where $K_J$ is the number of observed points of discontinuity for which $n_{\cdot k} > 0$, $\boldsymbol{r} := (r_1, \ldots, r_J)^T$, $r. := \sum_{j=1}^{J} r_j$, $n_{\cdot k} := \sum_{j=1}^{J} n_{jk}$, and $m_j \in \mathbb{Z}$ is the random size of group $j$.*

## 2.2 Exchangeable Partition Probability Function and Prediction Rule

Having the ECPF does not directly lead to the EPPF for the BNBP, as an EPPF describes the distribution of the exchangeable random partitions of the data groups whose sizes are observed rather than being random. To arrive at the EPPF, first we organize $\boldsymbol{z}$ into a random count matrix $\mathbf{N}_J \in \mathbb{Z}^{J \times K_J}$, whose $j$th row represents the partition of the $m_j$ data points into the $K_J$ shared exchangeable clusters and whose order of these $K_J$ nonzero columns is chosen uniformly at random from one of the $K_J!$ possible permutations, then we obtain a prior distribution on a BNBP random count matrix as

$$f(\mathbf{N}_J|\boldsymbol{r}, \gamma_0, c) = \frac{1}{K_J!} \prod_{j=1}^{J} \frac{m_j!}{\prod_{k=1}^{K_J} n_{jk}!} f(\boldsymbol{z}, \boldsymbol{m}|\boldsymbol{r}, \gamma_0, c)$$

$$= \frac{\gamma_0^{K_J} e^{-\gamma_0[\psi(c+r.)-\psi(c)]}}{K_J!} \prod_{k=1}^{K_J} \frac{\Gamma(n_{\cdot k})\Gamma(c+r.)}{\Gamma(c+n_{\cdot k}+r.)} \prod_{j=1}^{J} \frac{\Gamma(n_{jk}+r_j)}{n_{jk}!\Gamma(r_j)}. \qquad (7)$$

As described in detail in [17], although the matrix prior does not appear to be simple, direct calculation shows that this random count matrix has $K_J \sim \mathrm{Pois}\{\gamma_0[\psi(c+r.)-\psi(c)]\}$ independent and identically distributed (i.i.d.) columns that can be generated via

$$\boldsymbol{n}_{:k} \sim \mathrm{DirMult}(n_{\cdot k}, r_1, \ldots, r_J), \quad n_{\cdot k} \sim \mathrm{Digam}(r., c), \qquad (8)$$

where $\boldsymbol{n}_{:k} := (n_{1k}, \ldots, n_{Jk})^T$ is the count vector for the $k$th column (cluster), the Dirichlet-multinomial (DirMult) distribution [19] has PMF $\mathrm{DirMult}(\boldsymbol{n}_{:k}|n_{\cdot k}, \boldsymbol{r}) = \frac{n_{\cdot k}!}{\prod_{j=1}^{J} n_{jk}!} \frac{\Gamma(r.)}{\Gamma(n_{\cdot k}+r.)} \prod_{j=1}^{J} \frac{\Gamma(n_{jk}+r_j)}{\Gamma(r_j)}$, and the digamma distribution [20] has PMF $\mathrm{Digam}(n|r, c) = \frac{1}{\psi(c+r)-\psi(c)} \frac{\Gamma(r+n)\Gamma(c+r)}{n\Gamma(c+n+r)\Gamma(r)}$, where $n = 1, 2, \ldots$. Thus in the prior, the BNBP generates a Poisson random number of clusters, the size of each cluster follows a digamma distribution, and each cluster is further partitioned into the $J$ groups using a Dirichlet-multinomial distribution [17].

With both the ECPF and random count matrix prior governed by the BNBP, we are ready to derive both the EPPF and prediction rule, given in the following two Lemmas, with proofs in the Appendix.

**Lemma 3** (EPPF). *Let $\sum_{\sum_{k=1}^{K} \boldsymbol{n}_{:k}=\boldsymbol{m}}$ denote a summation over all sets of count vectors with $\sum_{k=1}^{K} \boldsymbol{n}_{:k} = \boldsymbol{m}$, where $m. = \sum_{j=1}^{J} m_j$ and $n_{\cdot k} \geq 1$. The group-size dependent exchangeable partition probability function (EPPF) governed by the BNBP can be expressed as*

$$f(\boldsymbol{z}|\boldsymbol{m}, \boldsymbol{r}, \gamma_0, c) = \frac{\frac{\gamma_0^{K_J}}{\prod_{j=1}^{J} m_j!} \prod_{k=1}^{K_J} \left[ \frac{\Gamma(n_{\cdot k})\Gamma(c+r.)}{\Gamma(c+n_{\cdot k}+r.)} \prod_{j=1}^{J} \frac{\Gamma(n_{jk}+r_j)}{\Gamma(r_j)} \right]}{\sum_{K'=1}^{m.} \frac{\gamma_0^{K'}}{K'!} \sum_{\sum_{k'=1}^{K'} \boldsymbol{n}_{:k'}=\boldsymbol{m}} \prod_{k'=1}^{K'} \frac{\Gamma(n_{\cdot k'})\Gamma(c+r.)}{\Gamma(c+n_{\cdot k'}+r.)} \prod_{j=1}^{J} \frac{\Gamma(n_{jk'}+r_j)}{n_{jk'}!\Gamma(r_j)}}, \qquad (9)$$

*which is a function of the cluster sizes $\{n_{jk}\}_{k=1,K_J}$, regardless of the orders of the indices $k$'s.*

Although the EPPF is fairly complicated, one may derive a simple prediction rule, as shown below, to simulate exchangeable random partitions of grouped data governed by this EPPF.

**Lemma 4** (Prediction Rule). *With $y^{-ji}$ representing the variable $y$ that excludes the contribution of $x_{ji}$, the prediction rule of the BNBP group size dependent model in (3) can be expressed as*

$$P(z_{ji} = k|\boldsymbol{z}^{-ji}, \boldsymbol{m}, \boldsymbol{r}, \gamma_0, c) \propto \begin{cases} \frac{n_{\cdot k}^{-ji}}{c+n_{\cdot k}^{-ji}+r.}(n_{jk}^{-ji} + r_j), & \text{for } k = 1, \ldots, K_J^{-ji}; \\ \frac{\gamma_0}{c+r.} r_j, & \text{if } k = K_J^{-ji} + 1. \end{cases} \qquad (10)$$

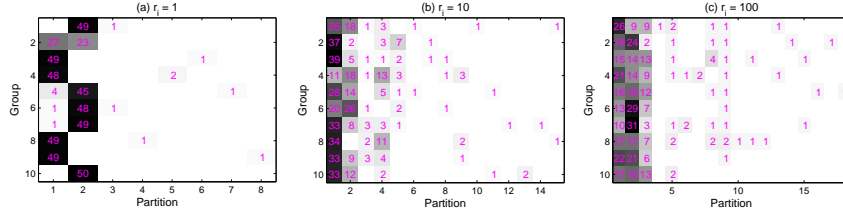

Figure 1: Random draws from the EPPF that governs the BNBP's exchangeable random partitions of 10 groups (rows), each of which has 50 data points. The parameters of the EPPF are set as $c = 2$, $\gamma_0 = \frac{12}{\psi(c+\sum_j r_j)-\psi(c)}$, and (a) $r_j = 1$, (b) $r_j = 10$, or (c) $r_j = 100$ for all the 10 groups. The $j$th row of each matrix, which sums to 50, represents the partition of the $m_j = 50$ data points of the $j$th group over a random number of exchangeable clusters, and the $k$th column of each matrix represents the $k$th nonempty cluster in order of appearance in Gibbs sampling (the empty clusters are deleted).

## 2.3 Simulating Exchangeable Random Partitions of Grouped Data

While the EPPF (9) is not simple, the prediction rule (10) clearly shows that the probability of selecting $k$ is proportional to the product of two terms: one is related to the $k$th cluster's overall popularity across groups, and the other is only related to the $k$th cluster's popularity at that group and that group's dispersion parameter; and the probability of creating a new cluster is related to $\gamma_0$, $c$, $r$, and $r_j$. The BNBP's exchangeable random partitions of the group-size count vector $\boldsymbol{m}$, whose prior distribution is governed by (9), can be easily simulated via Gibbs sampling using (10).

Running Gibbs sampling using (10) for 2500 iterations and displaying the last sample, we show in Figure 1 (a)-(c) three distinct exchangeable random partitions of the same group-size count vector, under three different parameter settings. It is clear that the dispersion parameters $\{r_j\}_j$ play a critical role in controlling how overdispersed the counts are: the smaller the $\{r_j\}_j$ are, the more overdispersed the counts in each row and those in each column are. This is unsurprising as in the BNBP's prior, we have $n_{jk} \sim \text{NB}(r_j, p_k)$ and $\boldsymbol{n}_{:k} \sim \text{DirMult}(n._k, r_1, \ldots, r_J)$. Figure 1 suggests that it is important to infer $r_j$ rather than setting them in a heuristic manner or fixing them.

## 3 Beta-Negative Binomial Process Topic Model

With the BNBP's EPPF derived, it becomes evident that the integer-valued BNBP also governs a prior distribution for exchangeable random partitions of grouped data. To demonstrate the use of the BNBP, we apply it to topic modeling [21] of a document corpus, a special case of mixture modeling of grouped data, where the words of the $j$th document $x_{j1}, \ldots, x_{jm_j}$ constitute a group $\boldsymbol{x}_j$ ($m_j$ words in document $j$), each word $x_{ji}$ is an exchangeable group member indexed by $v_{ji}$ in a vocabulary with $V$ unique terms. We choose the base distribution as a $V$ dimensional Dirichlet distribution as $g_0(\boldsymbol{\phi}) = \text{Dir}(\boldsymbol{\phi}; \eta, \ldots, \eta)$, and choose a multinomial distribution to link a word to a topic (atom). We express the hierarchical construction of the BNBP topic model as

$$x_{ji} \sim \text{Mult}(\boldsymbol{\phi}_{z_{ji}}), \ \boldsymbol{\phi}_k \sim \text{Dir}(\eta, \ldots, \eta), \ z_{ji} \sim \sum_{k=1}^{\infty} \frac{\theta_{jk}}{\Theta_j(\Omega)} \delta_k, \ m_j \sim \text{Pois}(\Theta_j(\Omega)),$$

$$\Theta_j \sim \Gamma P\left(r_j, \frac{B}{1-B}\right), \ r_j \sim \text{Gamma}(a_0, 1/b_0), \ B \sim \text{BP}(c, B_0), \ \gamma_0 \sim \text{Gamma}(e_0, 1/f_0). \quad (11)$$

Let $n_{vjk} := \sum_{i=1}^{m_j} \delta_v(x_{ji})\delta_k(z_{ji})$. Multiplying (4) and the data likelihood $f(\boldsymbol{x}_j|\boldsymbol{z}_j, \boldsymbol{\Phi}) = \prod_{v=1}^{V} \prod_{k=1}^{\infty} (\phi_{vk})^{n_{vjk}}$, where $\boldsymbol{\Phi} = (\boldsymbol{\phi}_1, \ldots, \boldsymbol{\phi}_\infty)$, we have $f(\boldsymbol{x}_j, \boldsymbol{z}_j, m_j|\boldsymbol{\Phi}, \Theta_j) = \frac{\prod_{k=1}^{\infty} \prod_{v=1}^{V} n_{vjk}!}{m_j!} \prod_{k=1}^{\infty} \prod_{v=1}^{V} \text{Pois}(n_{vjk}; \phi_{vk}\theta_{jk})$. Thus the BNBP topic model can also be considered as an infinite Poisson factor model [10], where the term-document word count matrix $(m_{vj})_{v=1:V, \ j=1:J}$ is factorized under the Poisson likelihood as $m_{vj} = \sum_{k=1}^{\infty} n_{vjk}$, $n_{vjk} \sim$ $\text{Pois}(\phi_{vk}\theta_{jk})$, whose likelihood $f(\{n_{vjk}\}_{v,k}|\boldsymbol{\Phi}, \Theta_j)$ is different from $f(\boldsymbol{x}_j, \boldsymbol{z}_j, m_j|\boldsymbol{\Phi}, \Theta_j)$ up to a multinomial coefficient.

The full conditional likelihood $f(\boldsymbol{x}, \boldsymbol{z}, \boldsymbol{m}|\boldsymbol{\Phi}, \boldsymbol{\Theta}) = \prod_{j=1}^{J} f(\boldsymbol{x}_j, \boldsymbol{z}_j, m_j|\boldsymbol{\Phi}, \Theta_j)$ can be further expressed as $f(\boldsymbol{x}, \boldsymbol{z}, \boldsymbol{m}|\boldsymbol{\Phi}, \boldsymbol{\Theta}) = \left\{\prod_{k=1}^{\infty} \prod_{v=1}^{V} \phi_{vk}^{n_{v\cdot k}}\right\} \cdot \left\{\frac{\prod_{k=1}^{\infty} \prod_{j=1}^{J} \theta_{jk}^{n_{jk}} e^{-\theta_{jk}}}{\prod_{j=1}^{J} m_j!}\right\}$, where the marginalization of $\boldsymbol{\Phi}$ from the first right-hand-side term is the product of Dirichlet-multinomial distributions and the second right-hand-side term leads to the ECPF. Thus we have a fully marginalized

likelihood as $f(\boldsymbol{x}, \boldsymbol{z}, \boldsymbol{m}|\gamma_0, c, \boldsymbol{r}) = f(\boldsymbol{z}, \boldsymbol{m}|\gamma_0, c, \boldsymbol{r}) \prod_{k=1}^{K_J} \left[ \frac{\Gamma(V\eta)}{\Gamma(n_{\cdot k}+V\eta)} \prod_{v=1}^{V} \frac{\Gamma(n_{v\cdot k}+\eta)}{\Gamma(\eta)} \right]$. Directly applying Bayes' rule to this fully marginalized likelihood, we construct a nonparametric Bayesian fully collapsed Gibbs sampler for the BNBP topic model as

$$P(z_{ji} = k|\boldsymbol{x}, \boldsymbol{z}^{-ji}, \gamma_0, \boldsymbol{m}, c, \boldsymbol{r}) \propto \begin{cases} \frac{\eta+n_{v_{ji}\cdot k}^{-ji}}{V\eta+n_{\cdot k}^{-ji}} \cdot \frac{n_{\cdot k}^{-ji}}{c+n_{\cdot k}^{-ji}+r.} \cdot (n_{jk}^{-ji} + r_j), & \text{for } k = 1, \ldots, K_J^{-ji}; \\ \frac{1}{V} \cdot \frac{\gamma_0}{c+r.} \cdot r_j, & \text{if } k = K_J^{-ji} + 1. \end{cases} \quad (12)$$

In the Appendix we include all the other closed-form Gibbs sampling update equations.

## 3.1 Comparison with Other Collapsed Gibbs Samplers

One may compare the collapsed Gibbs sampler of the BNBP topic model with that of latent Dirichlet allocation (LDA) [22], which, in our notation, can be expressed as

$$P(z_{ji} = k|\boldsymbol{x}, \boldsymbol{z}^{-ji}, \boldsymbol{m}, \alpha, K) \propto \frac{\eta+n_{v_{ji}\cdot k}^{-ji}}{V\eta+n_{\cdot k}^{-ji}} \cdot (n_{jk}^{-ji} + \alpha), \quad \text{for } k = 1, \ldots, K, \quad (13)$$

where the number of topics $K$ and the topic proportion Dirichlet smoothing parameter $\alpha$ are both tuning parameters. The BNBP topic model is a nonparametric Bayesian algorithm that removes the need to tune these parameters. One may also compare the BNBP topic model with the HDP-LDA [6, 23], whose direct assignment sampler in our notation can be expressed as

$$P(z_{ji} = k|\boldsymbol{x}, \boldsymbol{z}^{-ji}, \boldsymbol{m}, \alpha, \tilde{\boldsymbol{r}}) \propto \begin{cases} \frac{\eta+n_{v_{ji}\cdot k}^{-ji}}{V\eta+n_{\cdot k}^{-ji}} \cdot (n_{jk}^{-ji} + \alpha\tilde{r}_k), & \text{for } k = 1, \ldots, K_J^{-ji}; \\ \frac{1}{V} \cdot (\alpha\tilde{r}_*), & \text{if } k = K_J^{-ji} + 1; \end{cases} \quad (14)$$

where $\alpha$ is the concentration parameter for the group-specific Dirichlet processes $\widetilde{\Theta}_j \sim \text{DP}(\alpha, \widetilde{G})$, and $\tilde{r}_k = \widetilde{G}(\omega_k)$ and $\tilde{r}_* = \widetilde{G}(\Omega \backslash \mathcal{D}_J)$ are the measures of the globally shared Dirichlet process $\widetilde{G} \sim \text{DP}(\gamma_0, \widetilde{G}_0)$ over the observed points of discontinuity and absolutely continuous space, respectively.

Comparison between (14) and (12) shows that distinct from the HDP-LDA that combines a topic's global and local popularities in an additive manner as $(n_{jk}^{-ji} + \alpha\tilde{r}_k)$, the BNBP topic model combines them in a multiplicative manner as $\frac{n_{\cdot k}^{-ji}}{c+n_{\cdot k}^{-ji}+r.} \cdot (n_{jk}^{-ji} + r_j)$. This term can also be rewritten as the product of $n_{\cdot k}^{-ji}$ and $\frac{n_{jk}^{-ji}+r_j}{c+n_{\cdot k}^{-ji}+r.}$, the latter of which represents how much the $j$th document contributes to the overall popularity of the $k$th topic. Therefore, the BNBP and HDP-LDA have distinct mechanisms to automatically shrink the number of topics. Note that while the BNBP sampler in (12) is fully collapsed, the direct assignment sampler of the HDP-LDA in (14) is only partially collapsed as neither the globally shared Dirichlet process $\widetilde{G}$ nor the concentration parameter $\alpha$ are marginalized out. To derive a collapsed sampler for the HDP-LDA that marginalizes out $\widetilde{G}$ (but still not $\alpha$), one has to use the Chinese restaurant franchise [6], which has cumbersome book-keeping as each word is indirectly linked to its topic via a latent table index.

# 4 Example Results

We consider the JACM[1], PsyReview[2], and NIPS12[3] corpora, restricting the vocabulary to terms that occur in five or more documents. The JACM corpus includes 536 documents, with $V = 1539$ unique terms and 68,055 total word counts. The PsyReview corpus includes 1281 documents, with $V = 2566$ and 71,279 total word counts. The NIPS12 corpus includes 1740 documents, with $V = 13,649$ and 2,301,375 total word counts. To evaluate the BNBP topic model[4] and its performance relative to that of the HDP-LDA, which are both nonparametric Bayesian algorithms, we randomly choose 50%

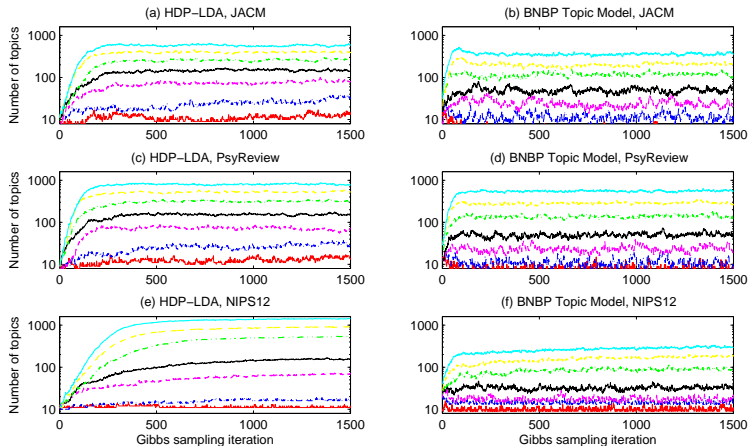

Figure 2: The inferred number of topics $K_J$ for the first 1500 Gibbs sampling iterations for the (a) HDP-LDA and (b) BNBP topic model on JACM. (c)-(d) and (e)-(f) are analogous plots to (a)-(c) for the PsyReview and NIPS12 corpora, respectively. From bottom to top in each plot, the red, blue, magenta, black, green, yellow, and cyan curves correspond to the results for $\eta = 0.50, 0.25, 0.10, 0.05, 0.02, 0.01$, and $0.005$, respectively.

of the words in each document as training, and use the remaining ones to calculate per-word heldout perplexity. We set the hyperparameters as $a_0 = b_0 = e_0 = f_0 = 0.01$. We consider 2500 Gibbs sampling iterations and collect the last 1500 samples. In each iteration, we randomize the ordering of the words. For each collected sample, we draw the topics $(\phi_k|-) \sim \mathrm{Dir}(\eta + n_{1 \cdot k}, \ldots, \eta + n_{J \cdot k})$, and the topics weights $(\theta_{jk}|-) \sim \mathrm{Gamma}(n_{jk} + r_j, p_k)$ for the BNBP and topic proportions $(\boldsymbol{\theta}_k|-) \sim \mathrm{Dir}(n_{j1} + \alpha \tilde{r}_1, \ldots, n_{jK_J} + \alpha \tilde{r}_{K_J})$ for the HDP, with which the per-word perplexity is computed as $\exp \left( - \frac{1}{m_{\cdot\cdot}^{\mathrm{test}}} \sum_v \sum_j m_{vj}^{\mathrm{test}} \ln \frac{\sum_s \sum_k \phi_{vk}^{(s)} \theta_{jk}^{(s)}}{\sum_s \sum_v \sum_k \phi_{vk}^{(s)} \theta_{jk}^{(s)}} \right)$, where $s \in \{1, \ldots, S\}$ is the index of a collected MCMC sample, $m_{vj}^{\mathrm{test}}$ is the number of test words at term $v$ in document $j$, and $m^{\mathrm{test}} = \sum_v \sum_j m_{vj}^{\mathrm{test}}$. The final results are averaged over five random training/testing partitions. The evaluation method is similar to those used in [24, 25, 26, 10]. Similar to [26, 10], we set the topic Dirichlet smoothing parameter as $\eta = 0.01, 0.02, 0.05, 0.10, 0.25$, or $0.50$. To test how the algorithms perform in more extreme settings, we also consider $\eta = 0.001, 0.002$, and $0.005$. All algorithms are implemented with unoptimized Matlab code. On a 3.4 GHz CPU, the fully collapsed Gibbs sampler of the BNBP topic model takes about 2.5 seconds per iteration on the NIPS12 corpus when the inferred number of topics is around 180. The direct assignment sampler of the HDP-LDA has comparable computational complexity when the inferred number of topics is similar. Note that when the inferred number of topics $K_J$ is large, the sparse computation technique for LDA [27, 28] may also be used to considerably speed up the sampling algorithm of the BNBP topic model.

We first diagnose the convergence and mixing of the collapsed Gibbs samplers for the HDP-LDA and BNBP topic model via the trace plots of their samples. The three plots in the left column of Figures 2 show that the HDP-LDA travels relatively slowly to the target distributions of the number of topics, often reaching them in more than 300 iterations, whereas the three plots in the right column show that the BNBP topic model travels quickly to the target distributions, usually reaching them in less than 100 iterations. Moreover, Figures 2 shows that the chains of the HDP-LDA are taking in small steps and do not traverse their distributions quickly, whereas the chains of the BNBP topic models mix very well locally and traverse their distributions relatively quickly.

A smaller topic Dirichlet smoothing parameter $\eta$ generally supports a larger number of topics, as shown in the left column of Figure 3, and hence often leads to lower perplexities, as shown in the middle column of Figure 3; however, an $\eta$ that is as small as $0.001$ (not commonly used in practice) may lead to more than a thousand topics and consequently overfit the corpus, which is particularly evident for the HDP-LDA on both the JACM and PsyReview corpora. Similar trends are also likely to be observed on the larger NIPS2012 corpus if we allow the values of $\eta$ to be even smaller than $0.001$. As shown in the middle column of Figure 3, for the same $\eta$, the BNBP topic model, usually representing the corpus with a smaller number of topics, often have higher perplexities than those of the HDP-LDA, which is unsurprising as the BNBP topic model has a multiplicative control mechanism to more strongly shrink the number of topics, whereas the HDP has a softer additive shrinkage mechanism. Similar performance differences have also been observed

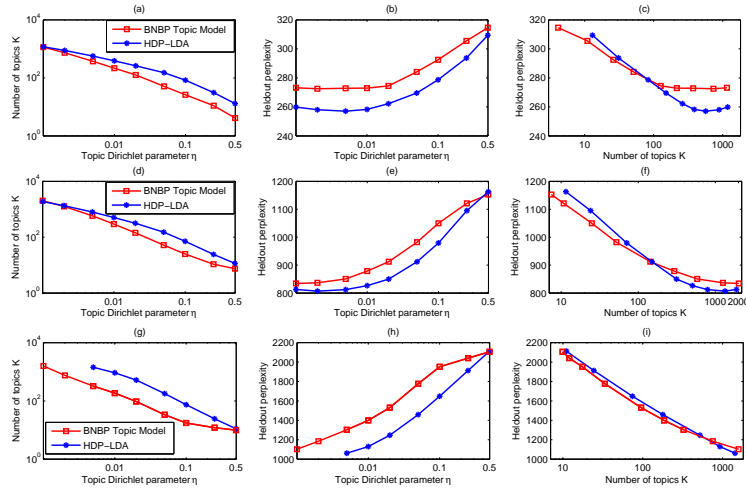

Figure 3: Comparison between the HDP-LDA and BNBP topic model with the topic Dirichlet smoothing parameter $\eta \in \{0.001, 0.002, 0.005, 0.01, 0.02, 0.05, 0.10, 0.25, 0.50\}$. For the JACM corpus: (a) the posterior mean of the inferred number of topics $K_J$ and (b) per-word heldout perplexity, both as a function of $\eta$, and (c) per-word heldout perplexity as a function of the inferred number of topics $K_J$; the topic Dirichlet smoothing parameter $\eta$ and the number of topics $K_J$ are displayed in the logarithmic scale. (d)-(f) Analogous plots to (a)-(c) for the PsyReview corpus. (g)-(i) Analogous plots to (a)-(c) for the NIPS12 corpus, where the results of $\eta = 0.002$ and $0.001$ for the HDP-LDA are omitted.

in [7], where the HDP and BNBP are inferred under finite approximations with truncated block Gibbs sampling. However, it does not necessarily mean that the HDP-LDA has better predictive performance than the BNBP topic model. In fact, as shown in the right column of Figure 3, the BNBP topic model's perplexity tends to be lower than that of the HDP-LDA if their inferred number of topics are comparable and the $\eta$ is not overly small, which implies that the BNBP topic model is able to achieve the same predictive power as the HDP-LDA, but with a more compact representation of the corpus under common experimental settings. While it is interesting to understand the ultimate potentials of the HDP-LDA and BNBP topic model for out-of-sample prediction by setting the $\eta$ to be very small, a moderate $\eta$ that supports a moderate number of topics is usually preferred in practice, for which the BNBP topic model could be a preferred choice over the HDP-LDA, as our experimental results on three different corpora all suggest that the BNBP topic model could achieve lower-perplexity using the same number of topics. To further understand why the BNBP topic model and HDP-LDA have distinct characteristics, one may view them from a count-modeling perspective [7] and examine how they differently control the relationship between the variances and means of the latent topic usage count vectors $\{(n_{1k}, \ldots, n_{Jk})\}_k$.

We also find that the BNBP collapsed Gibbs sampler clearly outperforms the blocked Gibbs sampler of [7] in terms of convergence speed, computational complexity and memory requirement. But a blocked Gibbs sampler based on finite truncation [7] or adaptive truncation [11] does have a clear advantage that it is easy to parallelize. The heuristics used to parallelize an HDP collapsed sampler [24] may also be modified to parallelize the proposed BNBP collapsed sampler.

## 5 Conclusions

A group size dependent exchangeable partition probability function (EPPF) for mixed-membership modeling is developed using the integer-valued beta-negative binomial process (BNBP). The exchangeable random partitions of grouped data, governed by the EPPF of the BNBP, are strongly influenced by the group-specific dispersion parameters. We construct a BNBP nonparametric Bayesian topic model that is distinct from existing ones, intuitive to interpret, and straightforward to implement. The fully collapsed Gibbs sampler converges fast, mixes well, and has state-of-the-art predictive performance when a compact representation of the corpus is desired. The method to derive the EPPF for the BNBP via a group size dependent model is unique, and it is of interest to further investigate whether this method can be generalized to derive new EPPFs for mixed-membership modeling that could be introduced by other integer-valued stochastic processes, including the gamma-Poisson and gamma-negative binomial processes.

## Footnotes

[1]http://www.cs.princeton.edu/~blei/downloads/

[2]http://psiexp.ss.uci.edu/research/programs_data/toolbox.htm

[3]http://www.cs.nyu.edu/~roweis/data.html

[4]Matlab code available in http://mingyuanzhou.github.io/

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
