[Supplementary Material · BNBP_Collapsed_v6_arXiv_appendix.pdf]

# Appendix: Beta-Negative Binomial Process and Exchangeable Random Partitions for Mixed-Membership Modeling

## Logbeta Process

Denoting a transformed representation of the beta process as $Q = -\sum_{k=1}^{\infty} \ln(1-p_k)\delta_{\omega_k}$, then for each $A \subset \Omega$, using the Lévy-Khintchine theorem and (1), the Laplace transform of the random variable $Q(A) = -\sum_{k:\omega_k \in A} \ln(1-p_k)$ can be expressed as

$$\mathbb{E}[e^{-sQ(A)}] \quad = \exp\left\{ \int_{[0,1]\times A} \left[ (1-p)^s - 1 \right] \nu(dpd\omega) \right\} = \exp\left\{ -B_0(A)\left[ \psi(c+s) - \psi(c) \right] \right\},$$

where $\psi(x) = \frac{\Gamma'(x)}{\Gamma(x)}$ is the digamma function with $\psi(c+s) - \psi(c) = \sum_{i=0}^{\infty}\left( \frac{1}{c+i} - \frac{1}{c+i+s} \right)$. Thus $Q(A)$ is an infinitely divisible random variable, which is defined as the logbeta random variable as $Q(A) \sim \mathrm{logBeta}(B_0(A), c)$. We further define the associated completely random measure as the logbeta process $Q \sim \mathrm{logBP}(B_0, c)$, with Lévy measure $\nu(dqd\omega) = \frac{e^{-qc}}{1-e^{-q}}dqB_0(d\omega)$. The logbeta random variable is found to be useful to derive closed-form Gibbs sampling update equations for model parameters, as shown below. We mention that the logbeta process presented here is the same as the beta-stacy process of [1].

## Proof for Lemma 2

By separating the atoms within the absolutely continuous space and the atoms with positive counts, the conditional likelihood of the BNBP group size dependent mixed-membership model, as shown in (5), can be rewritten as

$$f(\boldsymbol{z}, \boldsymbol{m}|\boldsymbol{r}, B) = \frac{1}{\prod_{j=1}^{J} m_j!}\left\{ \prod_{k:n_{\cdot k}=0}(1-p_k)^{r_\cdot} \right\} \cdot \left\{ \prod_{k:n_{\cdot k}>0} p_k^{n_{\cdot k}}(1-p_k)^{r_\cdot} \prod_{j=1}^{J}\frac{\Gamma(n_{jk}+r_j)}{\Gamma(r_j)} \right\}.$$

Let $\mathcal{D}_J := \{\omega_k\}_{k:n_{\cdot k}>0}$ denote the set of all observed atoms with positive counts, and let $K_J := |\mathcal{D}_J|$ denote its cardinality. Our goal is to marginalize out the beta process $B$ to obtain the joint distribution of the cluster assignments $\boldsymbol{z}$ and the group-size vector $\boldsymbol{m}$. Fixing an arbitrary labeling of the atoms in $\mathcal{D}_J$ from 1 to $K_J$, we may further rewrite the joint conditional likelihood as

$$f(\boldsymbol{z}, \boldsymbol{m}|\boldsymbol{r}, B) = \frac{1}{\prod_{j=1}^{J}m_j!}e^{-Q(\Omega\backslash\mathcal{D}_J)r_\cdot}\prod_{k=1}^{K_J}\left[ p_k^{n_{\cdot k}}(1-p_k)^{r_\cdot}\prod_{j=1}^{J}\frac{\Gamma(n_{jk}+r_j)}{\Gamma(r_j)} \right], \qquad (15)$$

where $Q(\Omega\backslash\mathcal{D}_J) := -\sum_{k:n_k=0}\ln(1-p_k)$ follows the $\mathrm{logBeta}(\gamma_0, c)$ distribution in the prior. Since $\int_{[0,1]\times\Omega}p^n(1-p)^r\nu(dpd\omega) = \gamma_0\frac{\Gamma(n)\Gamma(c+r)}{\Gamma(c+n+r)}$ and $\mathbb{E}_B[e^{-Q(\Omega\backslash\mathcal{D}_J)r_\cdot}] = e^{-\gamma_0[\psi(c+r_\cdot)-\psi(c)]}$, we may marginalize $B$ out of (15) with the Palm formula [2, 3, 4], leading to (6), which is a PMF that is only related to the cluster sizes, regardless of their orders. Since the group sizes $\{m_j\}_j$ themselves are random, and the random cluster sizes $\{n_{jk}\}_k$ are exchangeable, we call (6) as the exchangeable cluster probability function (ECPF) of the BNBP group size dependent mixed-membership model. □

## Proof for Lemma 3

As the group-size count vector $\boldsymbol{m} = (m_1, \ldots, m_J)^T$ can be generated as the summation of a Poisson random number of i.i.d. random count vectors, its PMF can be expressed as

$$f(\boldsymbol{m}|\boldsymbol{r}, \gamma_0, c) = \sum_{K=1}^{m_\cdot}\mathrm{Pois}\{K; \gamma_0\left[\psi(c+r_\cdot) - \psi(c)\right]\}\sum_{\sum_{k=1}^{K}\boldsymbol{n}_{:k}=\boldsymbol{m}}\prod_{k=1}^{K}\mathrm{DirMult}(\boldsymbol{n}_{\cdot k}|n_{\cdot k}, \boldsymbol{r})\mathrm{Digam}(n_{\cdot k}|r_\cdot, c)$$

$$= \sum_{K=1}^{m_\cdot}\frac{\gamma_0^K e^{-\gamma_0[\psi(c+r_\cdot)-\psi(c)]}}{K!}\sum_{\sum_{k=1}^{K}\boldsymbol{n}_{:k}=\boldsymbol{m}}\prod_{k=1}^{K}\frac{\Gamma(n_{\cdot k})\Gamma(c+r_\cdot)}{\Gamma(c+n_{\cdot k}+r_\cdot)}\prod_{j=1}^{J}\frac{\Gamma(n_{jk}+r_j)}{n_{jk}!\Gamma(r_j)}.$$

Using the ECPF in (6) and the multivariate distribution of the group size vector $\boldsymbol{m}$ shown above, the EPPF in (9) directly follows Bayes' rule as

$$f(\boldsymbol{z}|\boldsymbol{m}, \boldsymbol{r}, \gamma_0, c) = \frac{f(\boldsymbol{z}, \boldsymbol{m}|\boldsymbol{r}, \gamma_0, c)}{f(\boldsymbol{m}|\boldsymbol{r}, \gamma_0, c)}.$$

□

**Proof for Lemma 4**

One may rewrite the ECPF in (6) as

$$f(z_{ji}, \boldsymbol{z}^{-ji}, \boldsymbol{m}|\boldsymbol{r}, \gamma_0, c) = \frac{1}{\prod_{j=1}^{J} m_j!} \gamma_0^{K_J^{-ji}} e^{-\gamma_0[\psi(c+r.)-\psi(c)]} \left(\frac{\gamma_0 r_j}{c+r.}\right)^{\delta_{(K_J^{-ji}+1)}(z_{ji})}$$

$$\times \prod_{k=1}^{K_J^{-ji}} \left[ \frac{\Gamma(n_{.k}^{-ji} + \delta_k(z_{ji}))\Gamma(c+r.)}{\Gamma(c + n_{.k}^{-ji} + \delta_k(z_{ji}) + r.)} \prod_j \frac{\Gamma(n_{jk}^{-ji} + \delta_k(z_{ji}), +r_j)}{\Gamma(r_j)} \right],$$

which directly leads to (10) via Bayes' rule as

$$P(z_{ji}|\boldsymbol{z}^{-ji}, \boldsymbol{m}, \boldsymbol{r}, \gamma_0, c) = \frac{f(z_{ji}, \boldsymbol{z}^{-ji}, \boldsymbol{m}|\boldsymbol{r}, \gamma_0, c)}{\sum_{k=1}^{K_J^{-ji}+1} f(z_{ji} = k, \boldsymbol{z}^{-ji}, \boldsymbol{m}|\boldsymbol{r}, \gamma_0, c)}.$$

□

**Parameter Inference**

Using both the conditional likelihood (5) and marginal likelihood (6), with the data augmentation and marginalization techniques for the negative binomial distribution in [5, 6], we sample the model parameters as

$$(\gamma_0|-) \sim \text{Gamma}\left(e_0 + K_J, \frac{1}{f_0 + \psi(c+r.) - \psi(c)}\right), \tag{16}$$

$$(p_k|-) \sim \text{Beta}(n_{.k}, c+r.), \quad (Q(\Omega \backslash \mathcal{D}_J)|-) \sim \text{logBeta}(\gamma_0, c+r.), \tag{17}$$

$$(l_{jk}|-) = \sum_{t=1}^{n_{jk}} u_t, \ u_t \sim \text{Bernoulli}\left(\frac{r_j}{r_j + t - 1}\right), \tag{18}$$

$$(r_j|-) \sim \text{Gamma}\left(a_0 + \sum_{k=1}^{K_J} l_{jk}, \frac{1}{b_0 + Q(\Omega \backslash \mathcal{D}_J) - \sum_{k=1}^{K_J} \ln(1 - p_k)}\right). \tag{19}$$

To draw from the logBeta distribution $x \sim \text{logBeta}(\gamma_0, c+r.)$, we use its Laplace transform

$$\mathbb{E}[e^{-sx}] = \exp\left\{-\gamma_0\left[\psi(c+r. + s) - \psi(c+r.)\right]\right\}$$

together with the random number generating technique developed in [7]. The only parameter that we could not find an analytic conditional posterior is the concentration parameter $c$, for which we use the griddy-Gibbs sampler [8] to sample from a discrete distribution

$$(c|-) \propto f(\boldsymbol{z}, m|\boldsymbol{r}, \gamma_0, c) \tag{20}$$

over a grid of points $\frac{1}{1+c} = 0.01, 0.02, \ldots, 0.99$. Collapsed Gibbs sampling for the BNBP topic model is implemented by iteratively running (12) and (16)-(20). The direct assignment Gibbs sampler for the HDP-LDA is developed in [9] and also described in detail in [10].