[Reviews · NeurIPS 2014]

Submitted by Assigned_Reviewer_11

This paper presents a new Bayesian nonparametric model for mixed-membership modeling of grouped data. The model is based on a Beta-Negative Binomial Process (BNBP). Whereas this stochastic process was exploited by previous work, as mentioned in the paper, there are several aspects that set this work apart.

First, this work obtains an EPPF (exchangeable partition probability function), and derives the prediction rules thereon. The relations between Bayesian nonparametrics and EPPF has been studied for quite a while in statistics, following the seminal work by J. Pitman. Yet, this concept has not been extensively leveraged in the practice of BNP research. The connection between BNBP and EPPF made in this paper, to me, represents an important step along the direction of unifying the EPPF analysis and BNP research, which would potentially lead to more work along this line.

Second, the idea of randomizing the group sizes (which are treated as given constants in most other work) is interesting. It turns out that this idea makes it possible to derive some marginal distributions.

I think this paper has important theoretical contributions. However, the practical utility of this work remains in question. The inference algorithm is obviously more complex than HDP (in both theory and algorithms). The improvement demonstrated in the results is not sufficient to convince me that this increased complexity is worthy in practice. Also, the paper did not provide an intuitive explanation (or justification) as to why the proposed framework would actually work better than HDP in practice.
Summary: I think this paper has important theoretical value. Yet, the practical utility remains unclear. I do wish the authors can clarify how this method may improve the practical performance as opposed to HDP and why.

Overall, I still lean towards acceptance.

Submitted by Assigned_Reviewer_16

This paper proposed fully collapsed Gibbs (CG) sampler for the BNBP.
Lemma 1 and the derived sampler are novel and interesting.

However, my concern is the experiments.
Your motivation is to construct the CG sampler for the BNBP because of its simple implementation, fast convergence, good mixing, and state-of-the-art predictive performance.
That is, this work focuses on the sampling algorithm, not statistical modeling.
Therefore, you have to compare other sampling algorithms for the BNBP with the proposed algorithm, not HDP-LDA.
Summary: A fully collapsed Gibbs sampler for the BNBP was provided.
The experimental results of other sampling algorithms for the BNBP are needed .

Submitted by Assigned_Reviewer_27

The paper presents the exchangeable partition probability function (EPPF) for the beta negative binomial process (BNBP) and utilizes it to develop a collapsed Gibbs sampler for the BNBP topic model.

This is a well written, reasonably clear, technically sound paper. The primary result of the paper -- a derivation of the BNBP's EPPF for mixed-membership modeling (arrived at via "count-mixture" models) is both interesting in its own right and provides potentially useful tools for deriving EPPFs for related models.

Additional Comments:

While the BNBP topic model is a reasonable illustration of the utility of the developed EPPF and corresponding prediction rule, the improvements over the HDP-LDA are only marginal and the only confident conclusion one can draw is that the BNBP topic model and the HDP-LDA have distinct characteristics. A more interesting experiment would be to contrast the collapsed sampler against the sliced sampler of [10] and the truncated sampler of [8] for inference in the BNBP topic model. The uncollapsed representations have natural advantages when it comes to parallelization and it would be useful to examine mixing tradeoffs wrt to the collapsed sampler developed here.
Summary: Overall, this is an interesting paper and passes the publication threshold. I only have a few minor comments on the experimental section ( see above).
Author Feedback
Author rebuttal: Reviewer 11:
We appreciate your positive feedbacks on the paper’s theoretical contributions.

We strongly agree that it is important to unify the EPPF analysis and BNP research. Generalizing Pitman’s seminal work on the partition of a single set to the partition of multiple sets is critical for us to derive the fully collapsed inference. In addition, we also note that Pitman’s work focused on "infinite" EPPFs whose random partitions could be sequentially constructed. Subject to the constraint of sampling consistency, one has P(z_1,...,z_m|theta)=P(z_1|theta)P(z_2|z_1,theta)...P(z_m|z_1,...,z_{m-1},theta) for an infinite EPPF. Our EPPF definition is already more general in the sense that it is for multiple groups. Moreover, if we let $J=1$, we actually recover an EPPF for a single set. This EPPF by itself is also interesting as it is not subject to the sampling consistency constraint and could be considered as a "finite" EPPF in Pitman’s terminology [5].

The idea of randomizing the group sizes provides a blueprint to derive an EPPF for an integer-valued stochastic process: (1) derive the ECPF, (2) derive the marginal distribution of the group sizes, (3) derive the EPPF using Bayes rule.

Below we clarify how our method may improve the practical performance as opposed to HDP and why:

First, the proposed BNBP sampler in Eq (12) is a fully collapsed sampler, but at the same time is as easy to implement as the (partially collapsed) direct assignment sampler of the HDP in Eq (14). Both samplers have very similar computational complexity and memory requirement. The fully collapsed sampler of the HDP has to introduce an additional table index for each word and hence has much more complicated book-keeping [6]. In addition, this sampler, to our knowledge, was derived from the Chinese restaurant franchise analogy, but not from an explicitly expressed EPPF.

Second, we described in Lines 406-418 that the BNBP is preferred to the HDP when a compact latent representation is desired, according to our experimental results. We provided in Lines 299-304 some explanations (multiplicative v.s. additive interactions of different layers). More details about the advantages of the BNBP can be found in [8,10] and hence not repeated here due to space constraint. E.g., [8] argued that while rarely and lightly used topics are allowed in HDP-LDA, they are strongly discouraged in a BNBP topic model.

In summary, we believe a fully collapsed sampler (usually with fast convergence and good mixing) with simple implementation would be of great interest to topic modelers. Also for a practitioner, the provided closed-form Gibbs update equations could be very helpful.

Reviewer 16:
Thanks for your concern on experiments.

We chose to directly compare the proposed BNBP sampler with the HDP direct assignment sampler because they are similar in terms of implementation and computational and memory requirements. As the HDP direct assignment sampler is widely used, we believe our well-controlled comparison is quite informative.

We agree it would be helpful if we could compare the proposed sampler with an uncollapsed one. We did not include that comparison due to space constraint and several other reasons:

(1) It is well-accepted in the topic modeling community that a collapsed sampler usually has much faster convergence and better mixing than a uncollapsed one.

(2) A collapsed sampler of the BNBP is an exact sampler, whereas a uncollapsed sampler of the BNBP often relies on truncating the beta process.

(3) A collapsed sampler does not have to explicitly sample the topics and topic weights and instantiate unused topics. Thus its computational complexity is often much lower.

To address your concern, we will include the following comment: our experiments show that the BNBP collapsed Gibbs sampler is far superior to the blocked Gibbs sampler of [8] in terms of convergence, mixing, computational complexity and memory requirement.

Reviewer 27:
Thank you for your positive feedbacks and we totally agree with you that this paper provides potentially useful tools for deriving EPPFs for related models.

The HDP-LDA is an excellent algorithm, but there are several advantages of the BNBP that we’d like to emphasize.

First, our experimental results show that if a compact representation (small to moderate number of inferred topics) is desired, the BNBP topic model is preferred, as it outperforms the HDP-LDA in this scenario given a similar number of inferred topics. We believe this has to do with how different layers of the model are interacted: multiplicative for the BNBP v.s. additive for the HDP. More details about the advantages of the BNBP can be found in [8,10] and hence not repeated here due to space constraint.

Second, the BNBP fully collapsed sampler is as easy to implement as the partially collapsed direct assignment sampler of the HDP. The fully collapsed sampler of the HDP via Chinese restaurant franchise has much more complicated book-keeping and has to infer an additional table index for each word, and is not derived from an explicitly expressed EPPF.

We agree that it would be interesting to contrast against uncollapsed samplers of [10] and [8]. We did not include that comparison due to space constraint and several other reasons listed in our responses to Reviewer 16. We like the fully collapsed BNBP sampler much better as we had found that in topic modeling, it had much lower computation, faster convergence and better mixing than the uncollapsed sampler of [8].

We totally agree that an uncollapsed sampler is easy to parallelize. We will try to add some comments about that. But we also notice that some popular parallel topic modeling algorithms, such as those in [24], were actually based on collapsed samplers, though some heuristics have to be used. We think similar heuristics maybe applied to develop parallel algorithms based on the BNBP collapsed sampler.